# Multimodal Treatment with Cognitive Behavioral Therapeutic Intervention Plus Bladder Treatment Is More Effective than Monotherapy for Patients with Interstitial Cystitis/Bladder Pain Syndrome—A Randomized Clinical Trial

**DOI:** 10.3390/jcm11206221

**Published:** 2022-10-21

**Authors:** Wan-Ru Yu, Jia-Fong Jhang, Bai-Yueh Chen, Syuan-Ru Ou, Hao-Ming Li, Hann-Chorng Kuo

**Affiliations:** 1Department of Nursing, Hualien Tzu Chi Hospital, Buddhist Tzu Chi Medical Foundation, Hualien 970, Taiwan; 2Institute of Medical Sciences, Tzu Chi University, Hualien 970, Taiwan; 3Department of Urology, Hualien Tzu Chi Hospital, Buddhist Tzu Chi Medical Foundation and Tzu Chi University, Hualien 970, Taiwan; 4Department of Psychiatry, Hualien Tzu Chi Hospital, Buddhist Tzu Chi Medical Foundation, Hualien 970, Taiwan

**Keywords:** Interstitial cystitis/bladder pain syndrome, cognitive behavioral therapy, anxiety severity, quality of life, treatment outcome, urinary symptoms, bladder treatment

## Abstract

(1) Background: Introduction: Interstitial cystitis/bladder pain syndrome (IC/BPS) not only induces physiological damage but also greatly affects psychological stress. Multidisciplinary therapy has been recommended for IC/BPS treatment, but clinical trial data of combined bladder therapy and cognitive behavioral therapy (CBT) are lacking. This study evaluated CBT efficacy in patients with IC/BPS. (2) Methods: Patients with IC/BPS were randomized to the bladder monotherapy (BT) or combined CBT (CBT) group. The primary endpoint was the self-reported outcome by global response assessment (GRA). Secondary endpoints included IC symptoms and problem index, bladder pain score, Beck’s anxiety inventory (BAI), and depression inventory, and objective parameters were also compared. (3) Result: A total of 30 patients receiving BT and 30 receiving CBT therapy were enrolled. Significant improvement of the BAI at 8 (*p* = 0.045) and 12 weeks (*p* = 0.02) post-treatment was observed in the CBT group, with significantly greater GRA scores at 12 weeks (*p* < 0.001). Repeated measures analysis of variance showed a significant effect within the CBT group on IC/BPS patients’ self-reported treatment outcomes (*p* = 0.001) and anxiety severity BAI scores (*p* = 0.033). (4) Conclusion: A multimodal treatment of CBT combined with suitable bladder treatment more effectively improves anxiety severity and treatment outcomes in patients with IC/BPS.

## 1. Introduction

Interstitial cystitis/bladder pain syndrome (IC/BPS) is defined as the presence of chronic pelvic pain, pressure, or discomfort perceived to be associated with the urinary bladder accompanied by at least one of the following urinary symptoms: frequency, nocturia, and urgency [1], for six weeks to six months [2,3]. The reported prevalence of IC/BPS ranges from 0.01% to 2.3%, and the Taiwan National Database showed that the prevalence was 0.04% in 2013 [1]. The pathogenesis of IC/BPS might be due to neurogenic inflammation which activates bladder-afferent nerves and provokes bladder pain and a hypersensitive bladder with increased apoptosis of urothelium [1,4]. Psychological stress had been shown to be associated with exacerbation of IC symptoms involving pain and inflammation, not only associated with physiological damage but also negatively affecting psychological stress, and leading to poor quality of life (QoL) due to the current absence of effective long-lasting treatment [4]. Appropriate treatment should be conducted hierarchically and should combine multiple approaches, starting from lifestyle modification, followed by bladder therapy, accompanied by psychological adjustment and coping support [2,4,5,6]. The goal of managing patients with IC/BPS is to provide symptom relief and help patients achieve acceptable QoL post-treatment [6].

Treatments for IC/BPS include conventional therapies and experimental therapies. The former category includes cystoscopic hydrodistention, oral analgesic and anti-inflammatory drugs, intravesical hyaluronic or chondrointin instillation, and intravesical Botox injections [1,2]. Experimental therapies include sacral nerve stimulation, intravesical steroid injection, low-energy shock-wave therapy, and intravesical platelet-rich plasma injection [7,8]. For IC patients with Hunner’s lesion, transurethral resection of the ulcer, and partial cystectomy with or without bladder augmentation might be necessary to eradicate the severe bladder pain and contracted bladder [1,2]. In real-life clinical practice, few patients with IC/BPS receive only a single treatment. In Asia, people often refuse to admit they might have psychological problems, due to traditional culture and their own embarrassment. These cultural values result in concealing pain, suffering, and anger, and prevent the expression of strong emotions, especially negative ones [9]. Nevertheless, IC/BPS is a condition characterized by combined physiological and psychological disorders; thus, adequate treatment requires not only therapy targeting the bladder but also multimodal therapy and behavioral modification.

Among the psychological symptoms, depression, anxiety, fatigue, and pain are psychological factors identified as risk factors for poor self-care, increased symptomatic burden, worsened physical functioning, more severe morbidity, and reduced QoL [10]. Cognitive behavioral therapy (CBT) is a formal psychological therapeutic approach that encourages self-management of illnesses using the biopsychosocial model [11]. Before 1976, Beck described cognitive therapy as a robust system of integrative psychotherapy, clearly demonstrating that a person’s view influences their assumptions, behaviors, and reactions to a certain situation [12].

Although psychological therapy has shown therapeutic benefits and has been helpful for patients with IC/BPS, data from randomized clinical trials investigating the efficacy of CBT are still lacking [13]. Therefore, this clinical trial aimed to provide the first evidence to evaluate the efficacy of CBT for the treatment of patients with IC/BPS.

## 2. Materials and Methods

### 2.1. Design and Ethical Considerations

Open-labeled convenience sampling was conducted in a medical center from January 2020 to December 2021. This study was guided by the CONSORT checklist [14], see Appendix A. This study was approved by the research ethics committee of Buddhist Tzu Chi General Hospital (IRB number: 108-212-A). The study conformed to the provisions of the Declaration of Helsinki. All patients provided written informed consent with confidentiality guarantees after receiving a complete description of the study protocol. 

### 2.2. Participants

Patients with IC/BPS were recruited from the Urology Outpatient Clinic. All participants were eligible if they met the ESSIC and AUA criteria for IC/BPS [1,3] and had a Beck’s anxiety inventory (BAI) score of >18 (indicating moderate or severe anxiety). These patients had already undergone cystoscopic hydrodistention and bladder biopsy to confirm the IC/BPS diagnosis [1].

Eligible patients of either sex had already been diagnosed with IC/BPS for at least 1 year, and were aged 20–80 years. All patients presented a chief complaint of bladder pain and frequency and had poor treatment outcomes after third-line or novel therapy for IC/BPS as described by the AUA guidelines. Patients had not changed their treatment regimen within the previous 12 weeks. Furthermore, patients had no urinary tract infection or overt neurogenic bladder dysfunction in the previous 6 months. Patients were excluded if they had mild anxiety severity or severe physical or mental illness, malignancy, infectious diseases in the lower urinary tract, or post-void residual volume of >200 mL.

### 2.3. Randomization

The investigators randomized participants 1:1 using simple randomization, to receive appropriate bladder treatment (BT Group) alone or 8-week CBT intervention in combination with an appropriate bladder treatment regimen (CBT group). Both patient groups received the same standardized specialty urological nursing guidance and education, including an educational handout, and regular follow-up following the IC/BPS health instruction booklet was conducted for all patients.

### 2.4. Treatment Procedure

The participants and providers were not masked to the intervention; for allocation of the participants, random sampling drawn by random numbers was used as a coordinator. Providers were instructed not to make changes to the IC/BPS bladder treatment regimens during the treatment period unless specifically requested by study participants due to worsening symptoms. The CBT was performed by a certified clinical psychologist with 19 years of clinical experience in psychological consultation. CBT sessions were conducted weekly, 50 min per session, for a total of 8 weeks. These sessions educated patients about relaxation training, mental health education, monitoring and regulation of emotions, positive reinforcement, cognitive and behavioral therapies, and other relaxation techniques simultaneously for generalization and enhancement (Figure 1).

### 2.5. Treatment Outcomes

The demographic data, baseline questionnaires, and clinical information including current and prior IC treatments were collected at enrolment. The primary endpoint was the self-reported treatment outcomes of CBT and BT Groups by the global response assessment (GRA) score, evaluated at 8 and 12 weeks after starting the treatment. The GRA was scored from −3 (markedly worse) to +3 (markedly improved) [15]. GRA scores of +2 or +3 were considered satisfactory treatment outcomes; all other GRA scores were considered unsatisfactory outcomes. The secondary endpoint included BAI and Beck’s depression severity index (BDI), IC symptoms index (ICSI), and IC problems index (ICPI) of O’Leary Sant scores (OSS), bladder pain severity on the numeric pain-rating scale (NRS), and uroflowmetry parameters evaluated at 8 and 12 weeks post-treatment. BAI scores of 0–18 points indicated mild anxiety, 19–29 points indicated moderate anxiety, and 30–63 indicated severe anxiety [16,17], whereas BDI scores of 0–13 indicated minimal depression, 14–19 indicated mild depression, 20–28 indicated moderate depression, and 29–63 indicated severe depression [18]. 

Before treatment, all patients underwent videourodynamic study (VUDS) and cystoscopic hydrodistention. The maximal bladder capacity (MBC), glomerulation grade after cystoscopic hydrodistention, and uroflowmetry parameters were recorded. After cystoscopic hydrodistention, patients were consecutively treated with bladder-targeting medications for bladder pain, including nonsteroidal anti-inflammatory drugs, cyclooxygenase-2 inhibitors, antimuscarinics, alpha-blockers, intravesical hyaluronic acid (HA) instillations, and intravesical botulinum toxin A (BoNT-A) injections [2] or novel treatment with plasma-rich platelet (PRP) injection [8].

Psychological assessments including Weng’s Taiwan Type-D scale (DS14) [19] and Chen’s perceived stress scale (PSS10) were also performed [19,20,21]. The DS14 responses including negative affectivity (NA) and social inhibition (SI) were indicated on a five-point Likert scale from 0 (false) to 4 (true), with scores range 0–28 for each subscale, a standardized cut-off of ≥10 on both subscales indicated type-D cases [22]. PSS was rated by a five-point response (0 = never, 1 = almost never, 2 = sometimes, 3 = fairly often, 4 = very often); scores ranged from 0–40, a high score indicating a high degree of perceived stress, and no cut-offs were predefined [23].

### 2.6. Sample Size and Statistical Analysis

The G*power version 3.1.9.6 calculator (Franz Faul, Universität Kiel, Germany) was used for the statistical tests followed by the repeated measures analysis of variance (ANOVA) between factors, with alpha error of 0.05, power of 0.80, number of measurements 3, and effect size 0.5 (medium) [13,24,25]. Statistical analysis was performed using statistical software SPSS version 25 (IBM, Armonk, NY, USA). A *p*-value of <0.05 was considered statistically significant. Between-group statistical comparisons were tested using Pearson’s chi-square test or Fisher’s exact test for categorical variables and an independent t test for continuous variables. Post-treatment parameters were compared between groups using the repeated measures ANOVA. Finally, univariate and multivariate logistic regression analyses were performed on all data.

## 3. Results

Between January 2020 and December 2021, 60 patients (5 men and 55 women) were enrolled and randomized to the BT and CBT groups, with 30 members in each. No patient dropped out, and one patient in the CBT Group was lost to follow-up at week 12. However, due to the coronavirus disease pandemic, six patients receiving CBT changed their attendance from face-to-face to online for the seventh and eighth sessions. Among all patients, 45 (75%) were married, mean IC/BPS duration was 9.1 years, 51 (85%) presented a chief complaint of perineal, bladder (73%), or urethral (17.9%) pain. The most frequently reported lower urinary tract symptoms were frequency (95%) and nocturia (73.2%). The mean MBC was 766.1 ± 192.4 mL, the mean glomerulation grade was 1.6 ± 0.9, and two patients had Hunner’s lesions. In the VUDS results, 15 patients (25%) revealed voiding dysfunction, 44 (73.3%) had a hypersensitive bladder, 12 (20%) had detrusor overactivity, and 47 (78.3%) had a positive potassium chloride (KCl) test with a painful response. In subjective assessments, PSS was 21.6 ± 7.1 points, DS14 was 31.3 ± 12.5 points, ICSI and ICPI indices were 13.9 ± 4.3 and 13.1 ± 3.2 points, respectively, the mean BAI was 27.1 ± 8.9, and the mean BDI was 27.9 ± 12.7 (Table 1).

We divided patients into two subgroups, the CBT and BT groups. Within the objective parameters, patients respectively received suitable treatment targeting the bladder pathology and clinical symptoms, including HA instillation, intravesical BoNT-A injection, and intravesical PRP injections. The patients’ characteristics at baseline showed no significant difference between groups, including objective parameters with age (*p* = 0.776), IC disease duration (*p* = 0.699), current treatment (*p* = 0.387), VUDS parameters, uroflowmetry parameters, MBC (*p* = 0.786), and glomerulation grade (*p* = 0.206) under cystoscopic hydrodistention, while even in repeated measures ANOVA also showed no significant differences in uroflowmetry parameters between groups at baseline or post-treatment. Likewise, baseline subjective parameters revealed no significant differences between groups for OSS, NRS, BAI, and BDI, including PSS and DS14. Furthermore, the subjective parameters of ICSI (*p* = 0.193), ICPI (*p* = 0.524), NRS (*p* = 0.597), BDI (*p* = 0.679), DS14 (*p* = 0.144), and PSS (*p* = 0.314) similarly showed no significant correlations between groups post-treatment (Table 2).

However, post-treatment anxiety according to BAI showed significant improvement at 8 and 12 weeks. Between-group changes also showed significant differences in BAI and GRA at 12 weeks. Repeated measures ANOVA was performed to compare the effects of CBT treatment combined with bladder treatment on IC/BPS patients’ treatment outcomes. Results of the repeated measures ANOVA showed a significant effect on self-reported treatment outcomes (F [2, 108] = 7.161, *p* = 0.001) and anxiety severity (F [2, 108] = 3.519, *p* = 0.033) within the CBT group. Scheffe’s post hoc tests showed that BAI severity was significantly lower in the CBT group compared with the BT Group (8 weeks: 18.6 ± 8.4 vs. 24.4 ± 12.7; 12 weeks: 14.8 ± 7.8 vs. 22.7 ± 10.6; *p* = 0.033) at 8 and 12 weeks post-treatment, respectively (Table 2).

## 4. Discussion

This study reveals that multimodal treatment including CBT combined with suitable bladder treatment was more effective than bladder treatment alone. The CBT intervention significantly improved subjective treatment outcomes and severity of anxiety in patients with IC/BPS with moderate anxiety refractory to conventional therapy. These patients received suitable treatments targeting bladder pathology along with symptomatic medication, education, and psychological support at every visit, even including support and encouragement from the IC/BPS support group on social media, which assisted them in regaining impetus for life.

To date, only two studies which included 21 patients with IC/BPS receiving mindfulness-based stress reduction intervention have been reported in the USA [13,26]. The current study was on Asian people and enrolled 30 patients receiving suitable bladder treatment combined with CBT intervention. The mean PSS10 score was 21.6 points for patients in this study, with higher perceived stress than the general public at approximately 11.9 to 12.6 points [20,23,27]. Negative affectivity and social inhibition of these patients on the DS14 scale were >20 points; suggesting that illness-related cognition affected illness-related coping behaviors, distress, symptom severity, and QoL [28]. In addition to bolstering physical health, mental health promotion is also essential in the multidisciplinary management of patients with IC/BPS.

CBT includes education, relaxation exercises, training in coping skills, stress management, and assertiveness. In cognitive therapy, the therapist helps the patient identify and correct distorted, maladaptive beliefs, whereas behavioral therapy uses thought exercises or real experiences to reduce symptoms and improve functioning [29]. Therefore, the overall symptoms in the CBT group were improved by alleviating some of the feelings of hopelessness associated with IC/BPS. It is also possible that CBT strategies provided coping mechanisms to deal with self-reported IC/BPS treatment outcomes and anxiety status, and improved patients’ anxiety levels and global response. Remarkably, existing evidence indicates changes in brain MRI images after CBT [30], and brain changes are also associated with greater pain, anxiety, and urological symptoms [31], all demonstrating an essential correlation between physiological and psychological disorders in patients with IC/BPS.

Although CBT intervention could affect treatment outcomes for patients with IC/BPS, no significant changes were observed in the objective parameters. However, using GRA, patients who received combined CBT reported significantly improved treatment results, suggesting that the impact of CBT on patients with IC/BPS was mainly on changes in mental stress rather than pathophysiological improvement of the bladder. Patients undergoing CBT had a mean MBC of 772 ± 182 mL and a mean glomerulation grade of 1.7 ± 1.0; however, their ICSI and ICPI were >27 points, with high perceived pressure and distress. These results further indicated that their bladder conditions were no worse than the other group, and severity of their IC symptoms was susceptible to environmental and mental stress, which varied with remissions and exacerbations.

Only two patients were diagnosed with Hunner’s lesion type IC/BPS in this trial. Although the treatment options for Hunner’s IC and non-Hunner’s IC are different, conservative treatment including behavior modification, stress reduction, dietary modification, and physiotherapy was provided as indicated in the program for patients with non-Hunner’s IC. Eventually, patients with Hunner’s IC presented advantages in symptom improvement after CBT, suggesting that CBT intervention is a significantly effective treatment for patients with all types of IC/BPS. This study provides strong evidence that combined CBT and BT might be a useful treatment for IC/BPS refractory to conventional therapy.

The strengths of this study include its randomized controlled trial design and the use of patient-centered, validated questionnaires as outcome measures. Limitations of this study include the small sample size and short follow-up time. Furthermore, the significant improvement in BAI and GRA in the intervention group might be due in part to greater attention paid to patients rather than anything specific in the CBT technique. In clinical practice, IC/BPS patients are typically around middle age and have to work or take care of a family. Their psychological stress is high because they have to play many roles in daily life, including those of partner, parent, or child. In the future, a thorough treatment strategy for IC/BPS should not only provide bladder-focused therapy and psychological treatment to patients, but also should extend care to their families to achieve better mental health in patients and those around them.

## 5. Conclusions

This study provides evidence to support the role of CBT in combination with conventional therapy administered to patients with IC/BPS, especially in patients with moderate to severe anxiety. Alongside treatment targeting pathophysiology of the bladder, CBT can provide education, behavioral modification, and support for coping with stress, increasing the tolerability of patients’ bladder conditions. CBT treatment should be considered as a complementary treatment modality to be incorporated into the care plan of patients with IC/BPS.

## Figures and Tables

**Figure 1 jcm-11-06221-f001:**
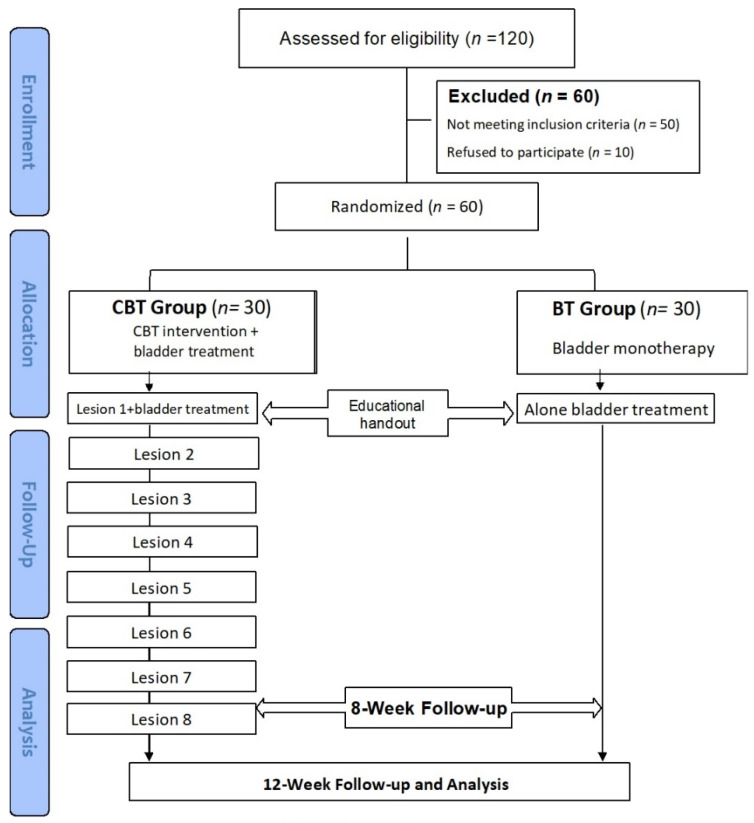
Study diagram.

**Table 1 jcm-11-06221-t001:** Baseline demographic data of patients with interstitial cystitis/bladder pain syndrome (*n* = 60).

Patient Characteristics	Variables and Descriptions	n (%)
Age (years)		53.5 ± 12.6
IC duration (years)		9.1 ± 7.4
ICSI		13.9 ± 4.3
ICPI		13.1 ± 3.2
Numerical rating pain scale		6.2 ± 2.7
Anxiety severity of BAI		27.1 ± 8.9
Depression severity of BDI		27.9 ± 12.7
PSS		21.6 ± 7.1
DS14		31.3 ± 12.5
VUDS parameters	First sensation of filling (mL)	117 ± 43
	Full sensation (mL)	199 ± 76
	Cystometric bladder capacity (mL)	236 ± 88
	Detrusor pressure (cm H_2_O)	22.4 ± 19.6
Uroflowmetry	Maximum flow rate (mL/s)	10.9 ± 4.9
	Voided volume (mL)	226 ± 112
	Post-void residual (mL)	36.1 ± 72.6
Maximal bladder capacity		766 ± 192
Glomerulation grade		1.6 ± 0.9
Voiding dysfunction	Bladder neck dysfunction	3 (5%)
	Dysfunction voiding	5 (8.3%)
	PRES	7 (11.6%)
Detrusor overactivity		12 (20%)
KCl test	No pain	5 (8.3%)
	Only pain	47 (78.3%)
	Only urge	5 (8.3%)
	Pain with urge	3 (5%)

ICSI, interstitial cystitis symptoms index; ICPI, interstitial cystitis problem index; BAI, Beck’s anxiety inventory; BDI, Beck’s depression inventory; PPS, perceived stress scale; DS14, type-D scale; VUDS, video urodynamic study; PRES, poor relaxation of the external sphincter.

**Table 2 jcm-11-06221-t002:** Comparison of objective and subjective parameters between the CBT intervention and control groups pre- and post-intervention (*n* = 60).

			CBT Group (*n =* 30)	BT Group (*n =* 30)	F	*p*-Value
Objective parameters	Age (years)		53.9 ± 11.9	53.0 ± 13.6		0.776
	IC duration (years)		9.5 ± 7.4	8.7 ± 7.5		0.699
	Current Treatment	HA instillation	0 (0%)	1 (3.3%)		0.387
		BoNT-A injection	3 (10%)	1 (3.3%)		
		PRP injection	27 (90%)	28 (93%)		
	VUDS parameters	FSF (mL)	113 ± 45.5	122 ± 40.2		0.414
		FS (mL)	187 ± 83.9	214 ± 64.4		0.186
		CBC (mL)	225 ± 98.4	250 ± 75.9		0.292
		Pdet (cm H_2_O)	24.9 ± 23.3	19.5 ± 14.2		0.309
	Qmax (mL/s)	Baseline	16.7 ± 8.8	15.3 ± 8.6	0.272	0.612
		8 weeks	17.4 ± 9.5	17.5 ± 8		
		12 weeks	18.7 ± 10.5	16.9 ± 11.3		
	Voided volume (mL)	Baseline	214 ± 115	222 ± 133	0.547	0.416
		8 weeks	193 ± 106	236 ± 141		
		12 weeks	218 ± 126	234 ± 130		
	PVR (mL)	Baseline	22.9.2 ± 23.9	14.5 ± 21.1	0.023	0.111
		8 weeks	25.2 ± 36.1	16.8 ± 18.9		
		12 weeks	23.1 ± 35	13.2 ± 14.2		
	MBC (mL)		772 ± 182	758 ± 206		0.786
	Glomerulation grade		1.7 ± 1	1.4 ± 0.7		0.206
Subjective parameters	ICSI	Baseline	13.9 ± 3.5	14 ± 5.2		0.193
		8 weeks	11.5 ± 4.2	10.5 ± 3.8		
		12 weeks	11 ± 3.8	10 ± 3.9		
	ICPI	Baseline	13.3 ± 2.4	12.9 ± 4		0.524
		8 weeks	11.9 ± 3.1	10.5 ± 3.8		
		12 weeks	11 ± 3.8	9.9 ± 3.4		
	OSS	Baseline	27.2 ± 5.4	26.9 ± 9	0.871	0.411
		8 weeks	23.5 ± 6.8	22.8 ± 7.8		
		12 weeks	22.7 ± 7.6	19.9 ± 6.7		
	NRS	Baseline	6 ± 2.2	6.4 ± 3.3	0.085	0.597
		8 weeks	4.6 ± 2.4	5 ± 2.8		
		12 weeks	4.3 ± 2.2	4.4 ± 2.8		
	BAI	Baseline	26.1 ± 8.2	28.3 ± 9.7	3.519	0.033 *
		8 weeks	18.6 ± 8.4 ^a^	24.4 ± 12.7 ^a^		
		12 weeks	14.8 ± 7.8 ^b.c^	22.7 ± 10.6 ^b.c^		
	BDI	Baseline	27.6 ± 13	28.2 ± 12.6	0.097	0.679
		8 weeks	21.8 ± 12.3	23.4 ± 11.8		
		12 weeks	21.2 ± 10.3	22.7 ± 11.6		
	GRA	8 weeks	1.3 ± 1.2	0.5 ± 1.6	7.161	0.001 *
		12 weeks	1.8 ± 0.7	0.5 ± 1.6		
	PPS		22.5 ± 7	20.6 ± 7.2		0.314
	DS14		33.5 ± 11.7	28.6 ± 13.1		0.144

HA, hyaluronic acid; BoNT-A, botulinum toxin A; PRP, plasma-rich platelet; FSF, first sensation of filling; FS, full sensation; CBC, cystometric bladder capacity; Pdet, detrusor pressure; Qmax, maximum flow rate; PVR, post-void residual; MBC, maximal bladder capacity; ICSI, interstitial cystitis symptoms index; ICPI, interstitial cystitis problem index; NRS, numerical rating pain scale; BAI, Beck’s anxiety inventory; BDI, Beck’s depression inventory; GRA, global response assessment; PPS, perceived pressure stress; DS14, type-D personality. * significant *p* < 0.05; a: 8 weeks between the CBT and BT groups, b: 12 weeks between the CBT and BT groups, c: each group between 12 and 8 weeks.

## Data Availability

To protect patient privacy and comply with relevant regulations, identified data are unavailable. Requests for de-identified data will be granted to qualified researchers with appropriate ethics board approvals and relevant data-use agreements.

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
