# Peer review of "Multimodal Treatment with Cognitive Behavioral Therapeutic Intervention Plus Bladder Treatment Is More Effective than Monotherapy for Patients with Interstitial Cystitis/Bladder Pain Syndrome—A Randomized Clinical Trial"

_jcm, 2022, doi:10.3390/jcm11206221_

Round 1

Reviewer 1 Report

Bladder pain syndrome/interstitial cystitis (BPS/IC) or primary bladder pain syndrome (PBPS) is a chronic non-infectious inflammation of the bladder that can generate supra-pubic, pelvic and/or abdominal pain for at least 6 months, frequency and urgency with incontinence. The diagnosis is given by the history and the exclusion of other pathologies both clinically or not and by cystoscopy and biopsy. Most patients improve by treatment, but healing is rare. Treatment varies and is often multimodal, including lifestyle changes, bladder gymnastics, analgesics, intravesical instillation and cognitive-behavioral therapies.

The incidence is not known, but the disorder appears to be more widespread than previously thought and it may be a warning of other clinically manifest systemic conditions. 

Caucasians and women are more likely to develop BPS/IC.

The etiopathogenesis is not known, but pathophysiology can imply the loss of protective urothelial mucin, with penetration of urinary K and other substances inside the bladder wall, activation of sensitive nerves and damage of smooth muscles.

Initially asymptomatic, the symptoms appear and worsen over the years: these worsen with bladder filling and decrease when patients urinate; in some patients, symptoms worsen during ovulation, menstruation, seasonal allergies, physical or emotional stresses or during sexual activity.

Although psychological therapy has shown beneficial effects on this condition, data are still lacking, so the aim of this work is to provide the first evidence of the efficacy of Cognitive-Behavioral Therapy (CBT) for the treatment of patients with BPS/IC.

Comment to Authors

Authors should be congratulated for the great work and the interesting topic discussed. 

This article emphasizes how crucial and pivotal multimodal therapy is to managing BPS/IC.

The manuscript needs a syntax check, tables are clearly described, and it is lacking in several points that would add value to the entire manuscript:

-      In the definition of BPS/IC the time criteria are missing. Please, correct it.

-      Please, authors should provide some epidemiological and physio-pathological data on the condition under examination.

-      Please, authors should provide more information about current therapy when they write about them. This work (doi: 10.3390/antibiotics10101194) can be taken as a reference for this purpose. It also emphasizes the importance of new therapeutic perspectives that would increase the scientific resonance of this paper. A reading is suggested. 

-      Materials and methods are robust.

Author Response

Dear Reviewers: Thank you for the constructive comments. We have revised the manuscript according to your suggestions. The followings are the point-to-point replies to the individual comment

.

Reviewer 1

Comments and Suggestions for Authors

Bladder pain syndrome/interstitial cystitis (BPS/IC) or primary bladder pain syndrome (PBPS) is a chronic non-infectious inflammation of the bladder that can generate supra-pubic, pelvic and/or abdominal pain for at least 6 months, frequency and urgency with incontinence. The diagnosis is given by the history and the exclusion of other pathologies both clinically or not and by cystoscopy and biopsy. Most patients improve by treatment, but healing is rare. Treatment varies and is often multimodal, including lifestyle changes, bladder gymnastics, analgesics, intravesical instillation and cognitive-behavioral therapies.

The incidence is not known, but the disorder appears to be more widespread than previously thought and it may be a warning of other clinically manifest systemic conditions.

Caucasians and women are more likely to develop BPS/IC.

The etiopathogenesis is not known, but pathophysiology can imply the loss of protective urothelial mucin, with penetration of urinary K and other substances inside the bladder wall, activation of sensitive nerves and damage of smooth muscles.

Initially asymptomatic, the symptoms appear and worsen over the years: these worsen with bladder filling and decrease when patients urinate; in some patients, symptoms worsen during ovulation, menstruation, seasonal allergies, physical or emotional stresses or during sexual activity.

Although psychological therapy has shown beneficial effects on this condition, data are still lacking, so the aim of this work is to provide the first evidence of the efficacy of Cognitive-Behavioral Therapy (CBT) for the treatment of patients with BPS/IC.

Comment to Authors

Authors should be congratulated for the great work and the interesting topic discussed.

This article emphasizes how crucial and pivotal multimodal therapy is to managing BPS/IC.

The manuscript needs a syntax check, tables are clearly described, and it is lacking in several points that would add value to the entire manuscript:

Point 1: In the definition of BPS/IC the time criteria are missing. Please, correct it

Response 1: Thank you for the comments. Refer to the American Urology Association (AUA) guideline requires the presence of symptoms for more than six weeks duration, and the ESSIC (International Society for the Study of IC/BPS) definition requires symptoms to be present for more than six months, whereas the International Continence Society and East Asian guidelines did not clearly define IC symptoms' duration time. In real-world practice, when patients visit our clinic usually have bladder pain and lower urinary symptoms already over 6 weeks, so we would follow the AUA and ESSIC guidelines to define the IC symptom duration of 6 weeks to six months (Line 38)

Point 2: Please, authors should provide some epidemiological and physio-pathological data on the condition under examination.

Response 2: Thank you for the comments. Referring to the latest East Asian IC guidelines in 2020, the known prevalence of IC or conditions suggestive of IC ranged from 0.01% to 2.3%, and in 2022 AUA guideline reported about 1.2 million women across the US had IC/BPS. The IC prevalence in Korea was reported to be 0.26% in women. The Taiwan National Database showed the prevalence was 0.04% in 2013, respectively. Actually, there is no objective marker to establish the presence of IC/BPS, and the studies to define real prevalence of IC/BPS are difficult to conduct. (Line 38-40) Although the etiology and pathogenesis of IC/BPS are still inconclusive, there is currently increasing support for the hypothesis of neurogenic inflammation which activates bladder-afferent nerves and provokes bladder pain and a hypersensitive bladder increased apoptosis of urothelium. We have added this statement in the Introduction. (Line 40-42)

Reference:

  1. Homma Y, Akiyama Y, Tomoe H, Furuta A, Ueda T, Maeda D, et al. Clinical guidelines for interstitial cystitis/bladder pain syndrome. International Journal of Urology 2020;27:578-89.
  2. Clemens JQ, Erickson DR, Varela NP, Lai HH. Diagnosis and Treatment of Interstitial Cystitis/Bladder Pain Syndrome. The Journal of Urology 2022;208:34-42.
  3. Chuang YC, Meng E, Chancellor M, Kuo HC. Pain reduction realized with extracorporeal shock wave therapy for the treatment of symptoms associated with interstitial cystitis/bladder pain syndrome—A prospective, multicenter, randomized, double‐blind, placebo‐controlled study. Neurourology and urodynamics 2020;39:1505-14.

Point 3: Please, authors should provide more information about current therapy when they write about them. This work (doi: 10.3390/antibiotics10101194) can be taken as a reference for this purpose. It also emphasizes the importance of new therapeutic perspectives that would increase the scientific resonance of this paper. A reading is suggested.

Response 3: Thank you for the comments. We have added the descriptions of current therapies for IC/BPS, and the recommended reference has been cited. This study not only focused on the pathophysiology of IC/BPS, but also on the therapeutic efficacy of combined psychological treatment with bladder-centered therapies. sure let us more enlightened for further research. (Lines  52-59)

Point 4: Materials and methods are robust.

Response 4: Thank you for the comments.

Reviewer 2 Report

The problem with patients of IC BPS is that by the time they reach a clinician who makes the right diagnosis, they have received many negative comments from previous clinicians as there is little or no evidence of disease on investigations. Hence not only do these patients suffer a 'Functional Overlay', the entire immediate  family, including spouse and children/parents  are suffering.  Therefore an important talk for the treating clinician is to  educate the immediate family in addition to treat the patient using CBT. 

Author Response

Dear Reviewers: Thank you for the constructive comments. We have revised the manuscript according to your suggestions. The followings are the point-to-point replies to the individual comment

Comments and Suggestions for Authors:

The problem with patients of IC/BPS is that by the time they reach a clinician who makes the right diagnosis, they have received many negative comments from previous clinicians as there is little or no evidence of disease on investigations. Hence not only do these patients suffer a 'Functional Overlay', the entire immediate family, including spouse and children/parents are suffering. Therefore, an important talk for the treating clinician is to educate the immediate family in addition to treat the patient using CBT.

Response: Thank you for the comments. Your positive comments inspire us for the future researches, not only to care the patients but also to attend their family. In clinical practice, these IC/BPS patients are around middle age and have to work or take care of a family. The psychological stress is high because they have to play many roles in daily life, including a partner, a parent, or a child. In the future, a thorough treatment strategy for IC/BPS should not only providing bladder-centered therapy and psychological treatment to patients, but also should extend the care to their families to achieve a better mental health in patients and their families. We have added this statement to the last paragraph of the discussion section. (Lines 292-297)

Round 2

Reviewer 1 Report

Thanks to the authors for the comprehensive comment reply. I am satisfied with the last paper version.